# MHA-RAG: Improving Efficiency, Accuracy, and Consistency by Encoding Exemplars as Soft Prompts

## Abstract

Adapting Foundation Models to new domains with limited training data is challenging and computationally expensive. While prior work has demonstrated the effectiveness of using domain-specific exemplars as in-context demonstrations, we investigate whether representing exemplars purely as text is the most efficient, effective, and stable approach. We explore an alternative: representing exemplars as soft prompts with an exemplar order invariant model architecture. To this end, we introduce Multi-Head Attention Retrieval-Augmented Generation (MHA-RAG), a framework with the number of attention heads serving as a simple hyperparameter to control soft prompt-generation across different tasks. Across multiple question-answering benchmarks and model scales, MHA-RAG achieves a 20-point performance gain over standard RAG, while cutting inference costs by a factor of $10\times$ GFLOPs—delivering both higher accuracy and greater efficiency, invariant to exemplar order.

## 1 Introduction

With the rapid scaling of model parameters, tuning Foundation Models for domain adaptation has become increasingly challenging due to both computational and data constraints. As an alternative, In-Context Learning (ICL) has emerged as a training-free adaptation strategy (Xie et al., 2021; Min et al., 2022). Rather than updating model parameters through gradient descent, ICL conditions the model at inference time by providing a small set of task-specific exemplars directly in the input prompt. Pioneering work such as GPT-3 (Brown et al., 2020) demonstrated that LLMs exhibit strong in-context learning capabilities when presented with exemplars in natural-language form. Subsequent studies in Wei et al. (2022) show that adding chain-of-thought exemplars further enhances performance by encouraging LLMs to generate intermediate reasoning steps. Remarkably, in certain settings, ICL has even outperformed fine-tuning-based methods (Mosbach et al., 2023; Pingua et al., 2025; Mallen et al., 2022).

Despite its promise, ICL faces three fundamental limitations. First, representing exemplars in textual format leads to **increasing inference cost**, because attention complexity quadratically grows with context length (Vaswani et al., 2017). Second, RAG has been shown to give **unsatisfiable performance** in domains with out-of-distribution data (Yu et al., 2024; Gupta et al., 2024). Third, ICL is highly sensitive to exemplar order (**exemplar-order variance**): prior studies demonstrate that reordering exemplars can cause substantial performance fluctuations (Pezeshkpour & Hruschka, 2023; Zheng et al., 2023), and we observe similar phenomenon in our own experiments (see Table 2).

To mitigate these issues, soft-prompt-based tuning is a viable choice. Instead of representing exemplars as long sequences of text, soft prompts encode exemplar information as a fixed set of trainable continuous vectors, which are prepended to the input. Because these vectors are much shorter than raw text, inference cost scales with the number of soft tokens rather than the full text length, greatly reducing quadratic overhead. While methods have been proposed for compressing text exemplars into soft prompts (Mu et al., 2023; Chevalier et al., 2023; Cheng et al., 2024; Rau et al., 2024; Li et al., 2024), these approaches often incur a noticeable performance degradation compared to text-based ICL. An alternative line of research investigates whether soft prompting can stand as an effective parameter-efficient fine-tuning method for domain adaptation (Lester et al., 2021; Liu et al.,

2024), (Liu et al., 2021). Approaches such as ATTEMPT (Asai et al., 2022), Instance-Dependent Prompt Generation (Wu et al., 2022) and LoPA (Jain et al., 2024a) demonstrate that adapting language models with soft prompts is a cost-saving mechanism that can enhance their performance without modifying model weights.

Therefore, in this paper, we ask the question: Can we learn a soft-prompt representation over in-context exemplars that simultaneously leverages 1) the cost efficiency of soft prompts, 2) the adaptability of in-context learning with retrieved exemplars, and 3) the architecture of neural networks that provide an exemplar-order invariant framework?

To address this question, we propose the Multi-Head Attention Retrieval-Augmented Generation framework (MHA-RAG). MHA-RAG learns to represent in-context exemplars as compact soft prompts and employs a multi-head scaled dot-product-based attention (Vaswani et al., 2017) to capture rich interactions between the query and each exemplar. Moreover, MHA-RAG introduces the number of heads as a tunable hyperparameter, enabling flexible control over the length of the generated soft prompts while simultaneously enforcing order-invariant aggregation across exemplar representations. Our experiments with a wide range of language models on domains like Chemistry and Medical QA demonstrate that this design provides (i) substantial gains in inference efficiency by reducing quadratic token overhead, (ii) performance improvements over text-based retrieval and prompt-tuning baselines, and (iii) stable and consistent results with the property of order-invariance embodied in the design.

In this work, we make the following contributions:

- We address the three fundamental limitations in RAG, which include high inference cost, unsatisfactory performance, and exemplar order variance within domain adaptions.

- We propose the MHA-RAG framework, which learns compact soft-prompt representations over in-context examples to reduce inference cost, while achieving high performance.

- We propose an exemplar order invariant multi-head-attention architecture to control soft-prompt generation in MHA-RAG, where the number of heads is a tunable hyperparameter.

- Our experiments show on average a 20-point performance gain over standard RAG across benchmarks and models, while cutting inference cost by $10\times$ GFLOPs.

## 2 RELATED WORK

### 2.1 PROMPT-COMPRESSION METHODS

Prompt-compression methods can be mainly divided into two categories: soft-prompt methods and hard-prompt methods. Soft-prompt methods compress long contexts into dense representations to reduce token count. GIST (Mu et al., 2023) learns "gist tokens" via modified attention masks, letting generation condition only on a compact set of vectors. AutoCompressor (Chevalier et al., 2023) and ICAE (Ge et al., 2023) similarly encode long inputs into summary vectors or memory slots that can be reused without re-encoding. More recent works like PISCO (Louis et al., 2025) and Embedding-based Memory Compressors (Dai et al., 2025) train memory tokens through distillation or pretraining, while CMT (Li et al., 2025) encodes documents into dense vectors via cross-attention. These methods achieve efficiency by replacing raw text with compact learned embeddings.

Hard-prompt methods remove tokens or text that contains less information. LLMLingua (Jiang et al., 2023) iteratively prunes tokens with low perplexity, while RECOMP (Xu et al., 2023) selects or summarizes sentences before prepending them to the input. Hierarchical methods, such as From Reading to Compressing (R2C) (Choi et al., 2024), compress at the token, chunk, and sentence levels, and training-free frameworks, like Perception Compressor (Tang et al., 2024), dynamically allocate compression ratios across different parts of the input. These techniques are often model-agnostic, but must balance compression rate with semantic fidelity.

In retrieval settings, compression enables handling many documents efficiently. COMPACT (Yoon et al., 2024) sequentially compresses document segments, while EXIT (Hwang et al., 2024) filters retrieved sentences to keep only relevant ones. xRAG (Cheng et al., 2024) projects a whole document into a single token embedding, and COCOM (Rau et al., 2024) pre-computes compact embeddings for

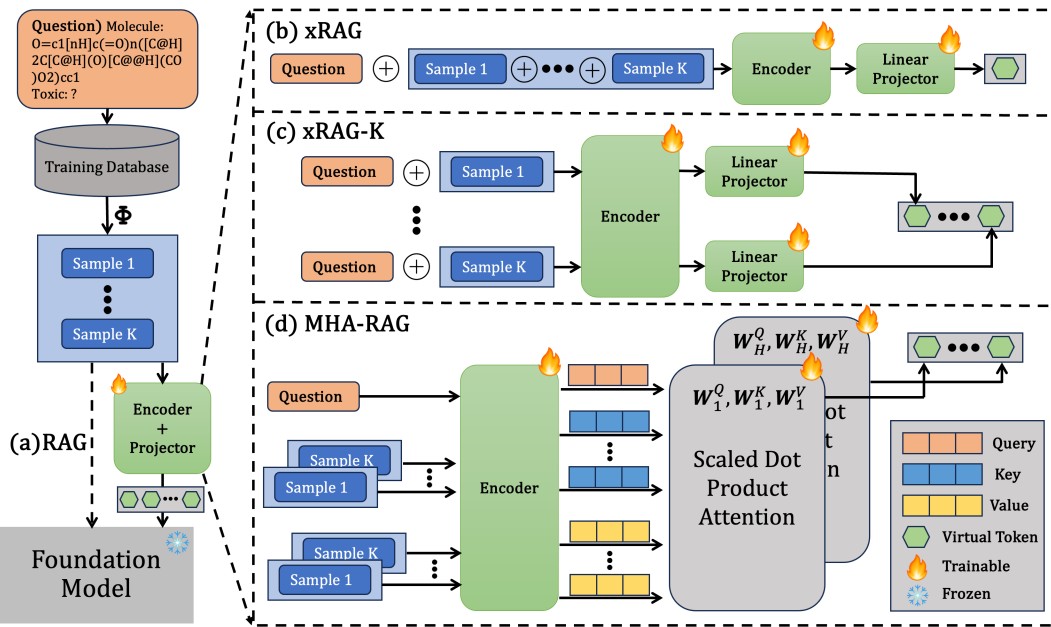

Figure 1: Comparison of domain-adaptation methods: (a) RAG uses retrieved exemplars directly as context, while (b) xRAG, (c) xRAG-K, and (d) MHA-RAG derive soft-prompt representations from exemplars. The figure illustrates how each method forms its representation (single vector in xRAG, $K$ vectors in xRAG-K, multi-head representations in MHA-RAG) and highlights their trainable components. Among the four methods, only MHA-RAG—due to its use of scaled dot-product attention—is invariant to the order of exemplars.

offline reuse. Beyond token inputs, DyPRAG (Tan et al., 2025) injects retrieved knowledge as LoRA weight updates, and AttentionRAG (Fang et al., 2025) uses model attention to prune retrievals in a training-free manner. Together, these methods reduce redundancy in RAG pipelines while controlling cost. Our MHA-RAG approach employs a multi-head attention-based soft prompt over retrieved in-context exemplars, enabling rich interactions between the query and exemplars, while ensuring that exemplar aggregation is order-invariant. Moreover, we can use the number of heads in the multi-head attention mechanism as a tunable hyperparameter to control soft-prompt generation across various tasks and models.

## 2.2 RETRIEVAL METHOD VERSUS FINE-TUNING

Retrieval-Augmented Generation (RAG) augments Foundation Models with external documents, providing grounded and up-to-date responses that surpass static fine-tuning methods in many knowledge-intensive tasks (Lewis et al., 2020). The proposed RAG model achieved state-of-the-art accuracy on open-domain QA benchmarks by retrieving different documents per query, outperforming purely fine-tuned parametric baselines. Similarly, REALM (Guu et al., 2020) and RETRO (Borgeaud et al., 2022) demonstrated that retrieval integration can enable smaller models to rival or surpass much larger fine-tuned ones.

Moreover, RAG leverages retrieved examples as in-context demonstrations, enabling Foundation Models to adapt to new domains without retraining. This strategy has been shown to improve performance in low-data and specialised settings, with unsupervised domain-adaptation approaches demonstrating significant gains in tasks like sentiment analysis and named entity recognition, etc. (Guu et al., 2020; Lewis et al., 2020; Borgeaud et al., 2022; Jain et al., 2024b). More recent benchmarks highlight that retrieval augmentation offers a lightweight, cost-efficient alternative to fine-tuning by dynamically expanding the model's accessible knowledge (Behrouz et al., 2024).

Another fundamental difference between RAG and fine-tuned models lies in test-time adaptability. RAG performs a fresh retrieval for each user query, dynamically tailoring the context to the question at hand. Recent work on dynamic test-time compute for LLMs explores strategies that allocate inference

resources adaptively based on query complexity. Snell et al. (2024) formalizes compute-optimal strategies—either using verifier-based search or adaptive response distribution, and outperforms much larger models under a fixed FLOPs budget. MHA-RAG leverages the benefits of RAG for domain adaptation—achieving superior performance in domain-specific tasks with less inference cost compared to RAG.

# 3 METHODOLOGY

The objective of domain adaptation is to adapt a language model $f_\theta$ to a new task with data $\mathbb{D} = \{(x_i, y_i)\}_{i=1}^N$ such that $y = f_\theta(x)$.

## 3.1 PRELIMINARIES

Formally, domain adaptation involves updating the parameters of the model $\theta$ to maximize the likelihood of the response $y$, i.e., $\max_\theta \mathbb{E}_{(x,y)\sim\mathbb{D}}[\log p_\theta(y|x)]$. In practice, Foundation Models are often adapted using parameter-efficient fine-tuning (PEFT) methods (He et al., 2021), where only a subset of parameters $\theta' \subset \theta$ is updated. This approach reduces training costs while preserving performance.

**Domain adaptation with in-context exemplars.** In domains where parameter updates are not feasible, either due to limited training data or training being computationally expensive, adaptation can be achieved by providing a set of a few domain-specific samples $\{(x_k, y_k)\}_{k=1}^K \subset \mathbb{D}$ as in-context exemplars. In this setting, the model prediction takes the form $y = f_\theta(\{(x_k, y_k)\}_{k=1}^K, x)$. This adaptation can be further enhanced by leveraging a relevance or similarity function $\Phi(x, x')$. Specifically, for each input $x$, the top-$K$ most relevant samples in $\mathbb{D}$ are retrieved as in-context exemplars—i.e., $(x_k, y_k) = \arg\max_{(x',y')\in\mathbb{D}\setminus\{(x,y),(x_1,y_1),...,(x_{k-1},y_{k-1})\}} \Phi(x, x')$, for $1 \le k \le K$.

**Representing in-context exemplars.** While approaches based on RAG typically represent in-context exemplars in text form, we instead consider soft prompts as an alternative representation. To this end, we define an encoding function $g(\mathbb{D}_K|x) = \mathbf{Z}$ that embeds $\mathbb{D}_K$, the top-$K$ in-domain samples retrieved for a given $x$, into a soft prompt $\mathbf{Z} \in \mathbb{R}^{d\times m}$, where $m$ denotes the virtual prompt length and $d$ is the hidden dimension of the language model. Similar to prompt tuning, the learned soft prompt is prepended to the word embeddings of $x$, thereby enabling domain adaptation without updating $\theta$.

## 3.2 MULTI-HEADED ATTENTION-BASED RAG

In this paper, we propose a multi-headed attention-based encoding function for RAG (MHA-RAG), where each head projects the top-$K$ in-context exemplars into the embedding space of the language model as a single soft-prompt vector $\mathbf{z} \in \mathcal{R}^d$. In particular, performing attention with $H$ heads results in the soft prompt $\mathbf{Z}_{MHA} = [\mathbf{z}^{(1)}, \dots \mathbf{z}^{(i)} \dots, \mathbf{z}^{(H)}]$ of length $m = H$, where the output from the $i^{th}$ head is

$$\mathbf{z}^{(i)} = \text{ATTENTION}(\mathbf{q}^i, \mathbf{K}^i, \mathbf{V}^i)$$
$$= \text{softmax}(\frac{\mathbf{q}^i \mathbf{K}^i}{\sqrt{d}})\mathbf{V}^i,$$
$$\text{with } \mathbf{K}^i = [\mathbf{k}_1^i, \dots \mathbf{k}_k^i \dots, \mathbf{k}_K^i] \text{ and } \mathbf{V}^i = [\mathbf{v}_1^i, \dots \mathbf{v}_k^i \dots, \mathbf{v}_K^i],$$

where $\mathbf{q}^i = \mathbf{W}_i^Q E_x$ is the attention query corresponding to a given $x$, and $\mathbf{k}_k^i = \mathbf{W}_i^K E_{x_k \oplus y_k}$ and $\mathbf{v}_k^i = \mathbf{W}_i^V E_{x_k \oplus y_k}$ are the keys and values corresponding to $(x_k, y_k) \in \mathbb{D}_K$, respectively.

$E_x \in \mathcal{R}^{d'}$ represents the dense representation of $x$ obtained by a sentence-embedding model with hidden dimension $d'$; $\oplus$ denotes the concatenation operator; and $\mathbf{W}_i^Q \in \mathcal{R}^{d\times d'}$, $\mathbf{W}_i^K \in \mathcal{R}^{d\times d'}$, and $\mathbf{W}_i^V \in \mathcal{R}^{d\times d'}$ represent the attention query, key, and value weights associated with head $i$, respectively. Overall, we optimize the following objective:

$$\max_\varphi \mathbb{E}_{(x,y)\sim\mathbb{D},\, \mathbb{D}_K=\Phi(x,\cdot)}\big[\log p(y|g_\varphi(\mathbb{D}_K|x), x)\big],$$

such that $g_\varphi(\mathbb{D}_K|x) = \mathbf{Z}_{MHA}$, where $\varphi$ represents the parameters of the encoding function $g$.

| Benchmarks | RAG | xRAG | xRAG-K | MHA-RAG |
|---|---|---|---|---|
| *Qwen3-0.6B* | | | | |
| BACE | 76.01 | 60.88$^{\downarrow -15.13}$ | 66.48$^{\downarrow -9.53}$ | 75.08$^{\downarrow -0.93}$ |
| BBBP | 66.62 | 68.16$^{\uparrow +1.54}$ | 76.57$^{\uparrow +9.95}$ | 87.82$^{\uparrow +21.20}$ |
| ClinTox | 53.30 | 42.54$^{\downarrow -10.76}$ | 66.05$^{\uparrow +12.75}$ | 97.12$^{\uparrow +43.82}$ |
| PubMedQA | 58.09 | 72.49$^{\uparrow +14.4}$ | 69.73$^{\uparrow +11.64}$ | 66.52$^{\uparrow +8.43}$ |
| *Qwen3-4B* | | | | |
| BACE | 75.87 | 59.07$^{\downarrow -16.8}$ | 55.46$^{\downarrow -20.41}$ | 76.27$^{\uparrow +0.4}$ |
| BBBP | 69.33 | 64.44$^{\downarrow -4.89}$ | 81.40$^{\uparrow +12.07}$ | 82.49$^{\uparrow +13.16}$ |
| ClinTox | 44.24 | 40.19$^{\downarrow -4.05}$ | 57.07$^{\uparrow +12.83}$ | 96.32$^{\uparrow +52.08}$ |
| PubMedQA | 79.26 | 71.16$^{\downarrow -8.10}$ | 70.36$^{\downarrow -8.9}$ | 76.59$^{\downarrow -2.67}$ |
| *Llama3.2-3B-Instruct* | | | | |
| BACE | 49.89 | 54.85$^{\downarrow -4.96}$ | 64.04$^{\uparrow +14.15}$ | 67.62$^{\uparrow +17.73}$ |
| BBBP | 46.15 | 72.87$^{\uparrow +26.72}$ | 84.09$^{\uparrow +37.94}$ | 87.61$^{\uparrow +41.46}$ |
| ClinTox | 37.90 | 62.88$^{\uparrow +24.98}$ | 86.89$^{\uparrow +48.99}$ | 94.44$^{\uparrow +56.54}$ |
| PubMedQA | 71.38 | 75.08$^{\uparrow +3.7}$ | 75.71$^{\uparrow +4.33}$ | 76.34$^{\uparrow +4.96}$ |
| $\Delta_{\text{RAG}}^{\text{avg}}$ | | -0.36 | 8.99 | 19.66 |

Table 1: Baseline comparison with RAG across benchmarks ($K = 5$). Performance is reported as effective accuracy—i.e, the geometric mean of the True-Positive and True-Negative rates ($\uparrow$: improvement relative to RAG, $\downarrow$: drop relative to RAG). Random guessing yields an effective accuracy of 50. Hyperparameters are tuned via sweeps: MHA-RAG ($H \in \{1, 2, 4, 8\}$). Results are averaged over 3 random seeds. Underlined indicates the best effective accuracy.

**Context compression and lower inference cost.** Prior work (Cheng et al., 2024) interprets the encoding function $g(\cdot)$ as a context-compression module achieving a compression ratio of $\frac{|\mathbb{D}_K|}{m}$. It reduces the tokenized length of the context $\mathbb{D}_K$ from $|\mathbb{D}_K|$ to $m$, thereby lowering overall inference cost during deployment. RAG without any compression yields a compression ratio of 1. In contrast, MHA-RAG achieves a higher and configurable compression ratio of $\frac{|\mathbb{D}_K|}{H}$ ($m = H$). Configurability comes from tailoring the soft-prompt size to the domain by varying the number of heads to adjust its representational capacity. In comparison, xRAG (Cheng et al., 2024), while achieving the highest possible compression (with $m = 1$), has a fixed ratio of $|\mathbb{D}_K|$ as it collapses the context into a single vector (see Figure 1 for reference).

**Order Invariance.** We argue that the soft prompts generated by MHA-RAG are invariant to the order of retrieved exemplars. This property follows from the use of scaled dot-product attention, which depends only on the set of input tokens rather than their order (Vaswani et al., 2017). Specifically, the inputs are exemplar embeddings $[E_{x_1 \oplus y_1}, \dots, E_{x_K \oplus y_K}]$, and permuting them does not alter the aggregated representation produced by each head. As a result, the generated soft prompt remains identical regardless of exemplar ordering. A proof of this property is provided in Appendix A.1.

## 4 ADAPTING TO QUESTION-ANSWERING DOMAINS

### 4.1 EXPERIMENTAL SETUP

**Benchmarks.** In this work, we focus on domains where language models exhibit weak zero-shot performance, but can benefit substantially from in-context exemplars or documents. Such exemplars provide cues in the form of analogies to similar samples, domain-specific formats (e.g., SMILES strings for molecules), specialized vocabulary and facts, or templates for step-by-step reasoning.

We therefore benchmark across two groups of tasks. The first group involves limited-data molecular-property-prediction tasks: (a) *BACE* [train/test split: 1413/100]—binary classification of whether a molecule inhibits BACE1; (b) *BBBP* [train/test split: 1950/100]—prediction of blood–brain barrier penetration (Yes/No); (c) *ClinTox* [train/test split: 1384/100]—classification of molecules as clinically

trial toxic vs. non-toxic (Guo et al., 2023). The second group consists of a medium-scale task: (d) *PubMedQA* [train/test split: 30250/890]—biomedical question answering (Jin et al., 2019).

**Performance Metrics.** For yes/no classification tasks, we report the geometric mean of the True-Positive and True-Negative rates, because it equally values both classes and avoids the biases of accuracy or F1 under skewed label distributions. We call this quantity the "Effective Accuracy." To quantify overall improvement relative to RAG, we compute $\Delta_{\text{RAG}}^{\text{avg}} = \sqrt[n]{\prod_{i=1}^{n}(100 + o_i^{method} - o_i^{RAG})} - 100$ where $o_i$ is the score of a method on task $i$, and $n$ is the total number of benchmark tasks across language models.

**Foundation Models** ($\theta$). We evaluate two families of models: *Llama-3.2-3B-Instruct* and the *Qwen3* series, specifically *Qwen3-0.6B* and *4B* (Yang et al., 2025).

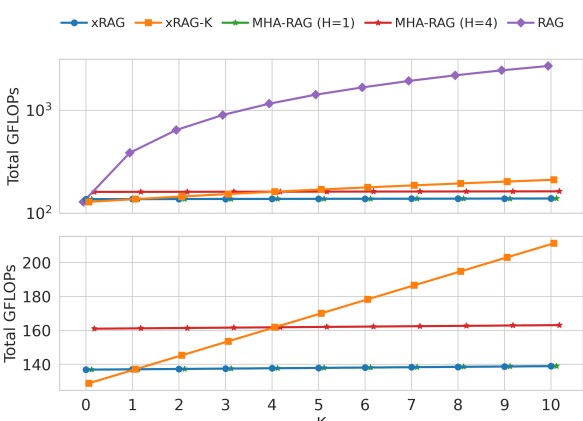

Figure 2: Comparison of inference compute in FLOPS with respect to $K$

**Sentence-Embedding Models** ($E_x$). To encode in-context exemplars, we employ domain-specific embedding models or encoder-only language models. For molecular tasks (*BACE1*, *BBBP*, *ClinTox*), we use *ChemBERTa-2-10M-MTR* (Ahmad et al., 2022). For biomedical QA (*PubMedQA*), we use *Qwen3-Embedding-0.6B* (Zhang et al., 2025).

**Retrieval Functions** ($\Phi$). We assume that a domain-specific retrieval function is provided. For molecular datasets, we retrieve the top-$K$ relevant exemplars using Tanimoto Similarity (Tanimoto, 1958; Guo et al., 2023), which computes scaffold-level similarity between SMILES representations.

For *PubMedQA*, we retrieve the top-$K$ documents most relevant to a question from the training corpus. Each document $doc_k$ is embedded into a dense vector $E_{doc_k}$ using *Qwen3-Embedding-8B* (Zhang et al., 2025), and retrieval is performed via cosine similarity.

**Baselines.** We compare against domain-adaptation baselines that do not fine-tune the foundation model. Instead, they leverage retrieved in-context exemplars from the training database. These include (a) *RAG*, which presents exemplars directly as text to the model, and (b) *xRAG* and (c) *xRAG-K*, which construct soft prompts from the exemplars. In *xRAG*, all exemplars are encoded into a single vector—i.e., $\mathbf{Z} = \text{MLP}(E_{x \oplus x_1 \oplus y_1 \dots x_K \oplus y_K})$, where $\mathbf{Z} \in \mathcal{R}^{d \times 1}$. *xRAG-K*, on the other hand, encodes each exemplar separately; i.e., $\mathbf{Z} = \text{MLP}(E_{x \oplus x_1 \oplus y_1}) \oplus \dots \oplus \text{MLP}(E_{x \oplus x_K \oplus y_K})$ where $\mathbf{Z} \in \mathcal{R}^{d \times K}$. Both methods employ a single-layer MLP to project the soft prompts from the embedding space of the sentence encoder into the base model's input space. Refer Figure 1.

**Training Details.** Models are tuned for 10 epochs on the limited-data molecular benchmarks (*BACE1*, *BBBP*, *ClinTox*) and 1 epoch on the larger-scale benchmarks (*PubMedQA*). Learning rates are selected from $\{1e-5, 3e-5, 5e-4\}$. The execution platform used 8× V100 GPUs (32GB VRAM each). The embedding models are fine-tuned with LoRA (rank 64). The comparison of overall trainable parameters are provided in Appendix A.5.

We report results on additional benchmarks and experiments in Appendix A.2 and A.3.

## 4.2 BASELINE COMPARISON

**Performance.** As shown by the underlining in Table 1, MHA-RAG achieves the greatest improvement in effective accuracy over RAG for almost all configurations tested, with an average gain of 19.66, computed as the geometric mean of improvements across tasks. This improvement stems from its flexible representational capacity: by varying the number of attention heads—each with its own set of weights—the model can specialize in capturing different types of dependencies across exemplars. In contrast, xRAG performs extreme compression by collapsing all retrieved samples into a single vector, which can possibly lead to loss of information. Its extension, xRAG-K improves over RAG

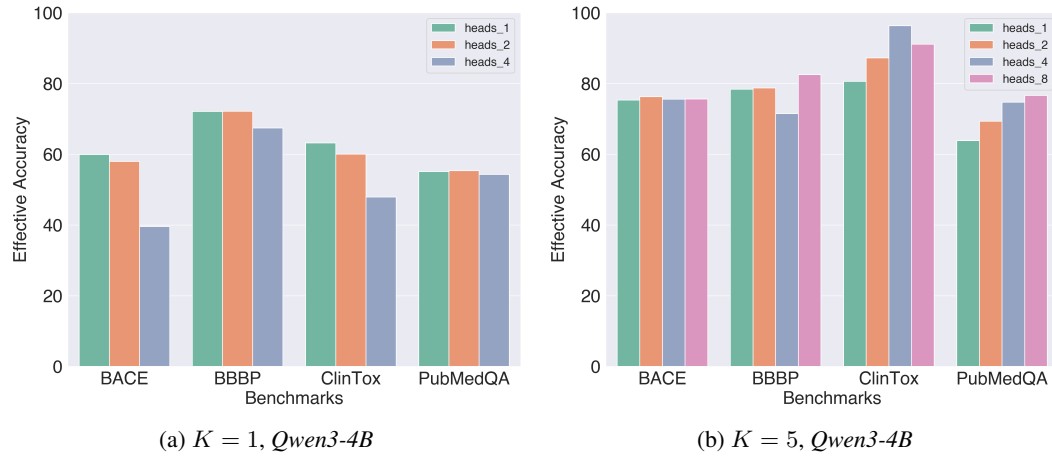

(a) $K = 1$, *Qwen3-4B*  (b) $K = 5$, *Qwen3-4B*

Figure 3: Varying number of heads in MHA-RAG. Performance averaged across 3 seeded runs.

by encoding each exemplar separately, but all exemplars are processed with identical weights. In contrast, MHA-RAG assigns distinct weights to each head and computes representations through attention over all exemplars, allowing head-specific representations.

**Inference Cost.** Figure 2 further shows that across $K = 1 \ldots 10$, RAG incurs substantially higher inference cost (in FLOPS) than MHA-RAG, due to its longer context length and the quadratic scaling of attention with respect to $K$. With a single head, MHA-RAG matches the cost of xRAG, and for head counts $< K$, it is more computationally efficient than xRAG-K.

**Order (In)variance.** From Table 2, we observe that all baselines—RAG, xRAG, and xRAG-K—exhibit non-zero variance when the order of in-context exemplars or documents is shuffled. For RAG, this sensitivity is expected because exemplars are concatenated as text, and order directly influences the model's input. In xRAG, order dependence arises from the positional encodings of the sentence-embedding encoder model: when all exemplars are jointly encoded into a single vector, reordering alters the resulting embedding. In xRAG-K, although each exemplar is encoded independently, the position of each resulting vector within the soft prompt is determined by exemplar order, leading to downstream variance. In contrast, our proposed MHA-RAG is order-invariant by design, as formally established in Appendix A.1.

| **Benchmarks** | RAG | xRAG | xRAG-K | Ours |
|---|---|---|---|---|
| BACE | 3.18 | 2.80 | 1.12 | **0.0** |
| BBBP | 2.25 | 3.02 | 1.27 | **0.0** |
| ClinTox | 8.52 | 0.12 | 4.91 | **0.0** |
| PubMedQA | 4.30 | 0.61 | 2.00 | **0.0** |

Table 2: Order-(in)variance analysis for *Qwen3-4B* with $K = 5$: standard deviation in performance when exemplar order is randomized across 5 seeded shuffles. (Lower numbers are better; zero means order-invariant.) **Bold** shows the lowest standard deviation achieved.

*Findings.* These results demonstrate that by spending an upfront cost for training, MHA-RAG achieves both higher effective accuracy than baselines and lower inference cost compared to RAG.

### 4.3 CONTEXT SATURATION WITH SOFT PROMPTS

In this section, we examine how effective accuracy varies with the number of retrieved exemplars ($K$), focusing on context saturation (Vladika & Matthes, 2025), the point at which adding more context introduces noise and causes effective accuracy to plateau or decline.

As shown in Table 3, increasing $K$ generally improves effective accuracy for both RAG ($K = 1 \rightarrow 5 \rightarrow 10$) and MHA-RAG ($K = 1 \rightarrow 5$), reflecting the benefit of richer context. A key finding is that MHA-RAG at K=5 generally outperforms RAG at both K=5 and K=10, indicating that it is more effective at extracting and representing information that is sufficient to answer the questions (Joren et al., 2024). Moreover, it achieves this performance with fewer FLOPs than RAG (See Figure 2).

| Benchmarks | RAG | | | MHA-RAG | | |
|---|---|---|---|---|---|---|
| | *K=1* | *K=5* | *K=10* | *K=1* | *K=5* | *K=10* |
| *Qwen3-4B* | | | | | | |
| BACE | 58.50 | 75.87 | 75.87 | 59.89 | **76.27** | 52.42 |
| BBBP | 67.50 | 69.33 | 68.01 | 72.12 | **82.49** | 80.43 |
| ClinTox | 30.94 | 44.24 | 54.48 | 63.19 | **96.32** | 89.44 |
| PubMedQA | 52.68 | 79.26 | **80.34** | 55.33 | 76.59 | 66.07 |
| *Llama3.2-3B-Instruct* | | | | | | |
| BACE | 51.71 | 49.89 | 36.36 | 52.37 | **67.62** | 54.73 |
| BBBP | 35.81 | 46.15 | 52.08 | 71.78 | **87.61** | 86.80 |
| ClinTox | 38.59 | 37.90 | 47.27 | 64.95 | **94.44** | 89.33 |
| PubMedQA | 54.59 | 71.38 | 73.53 | 57.91 | **76.34** | 72.50 |

Table 3: Effective accuracy of MHA-RAG vs. RAG under varying $K$. Results are averaged over 3 random seeds. **Bold** indicates the best performance. Underlined indicates the value of $K$ that achieves the best effective accuracy for a given method.

We further observe that MHA-RAG reaches context saturation earlier than RAG: effective accuracy peaks around K=5, whereas RAG continues to improve up to K=10. Beyond these points, additional exemplars degrade effective accuracy, likely due to the inclusion of less relevant or noisy samples, consistent with findings of Liu et al. (2023); Hsieh et al. (2024); Zhao (2023).

*Findings.* MHA-RAG enables more effective domain adaptation with fewer exemplars over RAG.

### 4.4 EFFECTIVE ACCURACY AS A FUNCTION OF ATTENTION HEADS

Having examined effective-accuracy variation with $K$, we now study the impact of the number of heads. From Figures 3a and 3b, we observe that when $K = 1$, increasing the number of heads does not improve effective accuracy, likely due to the limited context—only a single exemplar or document—offering little room for multiple heads to learn diverse representations. In contrast, at $K = 5$, increasing the number of heads generally leads to higher effective accuracy across benchmarks, suggesting that the benefit of multiple heads emerges only when sufficient context is available. Additional plots are provided in Appendix A.4.

*Findings.* MHA-RAG can effectively exploit additional context with an increasing number of heads.

### 4.5 COMPARISON WITH FINE-TUNING BASELINES

Because MHA-RAG requires some upfront training, we compare it against other fine-tuning methods. Specifically, we include LoRA (Hu et al., 2022) as a standard baseline for domain adaptation and evaluate against other soft-prompt fine-tuning methods—Prompt Tuning (PT) (Lester et al., 2021), Instance-Dependent Prompt Generation (IDPG) (Wu et al., 2022). As shown in Table 4, MHA-RAG achieves the highest average improvement of $47.45$ in effective-accuracy over LoRA, with gains computed as the geometric mean across tasks. We next explain the performance gaps method by method, focusing on one type of task at a time.

**Molecular-property-prediction tasks (BACE, BBBP, and ClinTox).** As shown in Table 4, LoRA often overfits on limited-data BACE, BBBP, and ClinTox benchmarks, collapsing to predictions dominated by a single class. Further, on these tasks MHA-RAG outperforms both PT and IDPG, achieving $\Delta_{\text{PT}}^{\text{avg}} = 69.95$ and $\Delta_{\text{IDPG}}^{\text{avg}} = 7.01$. This gap can be explained by how soft prompts are constructed: PT learns a fixed, question-independent prompt shared across the entire task, while IDPG conditions its generated prompt on the input question, but ignores related training exemplars. MHA-RAG instead conditions the soft prompt on retrieved exemplars that are most relevant to the current query, allowing it to further leverage task-specific information. These results highlight the benefit of grounding adaptation in limited data settings with retrieved exemplars, rather than relying solely on parametric tuning.

| Benchmarks | LoRA | Off-the-Shelf | PT | IDPG | MHA-RAG |
|---|---|---|---|---|---|
| *Qwen3-0.6B* | | | | | |
| BACE | 0.0 | $36.67^{\uparrow\ +36.67}$ | 0.0 | $53.75^{\uparrow\ +53.75}$ | $\underline{75.08}^{\uparrow\ +75.08}$ |
| BBBP | 0.0 | $22.21^{\uparrow\ +22.21}$ | 0.0 | $86.52^{\uparrow\ +86.52}$ | $\underline{87.82}^{\uparrow\ +87.82}$ |
| ClinTox | 0.0 | 0.0 | 0.0 | $94.87^{\uparrow\ +94.87}$ | $\underline{97.12}^{\uparrow\ +97.12}$ |
| PubMedQA | $\underline{73.94}$ | $13.96^{\downarrow\ -59.98}$ | $38.99^{\downarrow\ -34.95}$ | $45.50^{\downarrow\ -28.44}$ | $66.52^{\downarrow\ -7.42}$ |
| *Qwen3-4B* | | | | | |
| BACE | 12.32 | $0.0^{\downarrow\ -12.32}$ | $0.0^{\downarrow\ -12.32}$ | $62.54^{\uparrow\ 50.22}$ | $\underline{76.27}^{\uparrow\ -7.42}$ |
| BBBP | 19.61 | $10.88^{\downarrow\ -8.73}$ | $11.10^{\downarrow\ -8.51}$ | $\underline{87.14}^{\uparrow\ 67.53}$ | $82.49^{\uparrow\ 63.95}$ |
| ClinTox | 0.0 | 0.0 | 0.0 | $94.36^{\uparrow\ +94.36}$ | $\underline{96.32}^{\uparrow\ +96.32}$ |
| PubMedQA | 78.31 | $53.65^{\downarrow\ -24.66}$ | $81.04^{\uparrow\ +2.73}$ | $\underline{83.65}^{\uparrow\ +5.34}$ | $76.59^{\downarrow\ -1.72}$ |
| *Llama3.2-3B-Instruct* | | | | | |
| BACE | 0.0 | 0.0 | $13.61^{\uparrow\ +13.61}$ | $35.28^{\uparrow\ +35.28}$ | $\underline{67.62}^{\uparrow\ +67.62}$ |
| BBBP | 54.77 | $0.0^{\downarrow\ -54.77}$ | $36.89^{\downarrow\ -17.88}$ | $87.14^{\uparrow\ +32.37}$ | $\underline{87.61}^{\uparrow\ +32.84}$ |
| ClinTox | 31.62 | $0.0^{\downarrow\ -31.62}$ | $61.88^{\uparrow\ +30.26}$ | $\underline{94.87}^{\uparrow\ +63.25}$ | $94.44^{\uparrow\ +62.82}$ |
| PubMedQA | 82.15 | $56.73^{\downarrow\ -25.42}$ | $64.68^{\downarrow\ -17.47}$ | $\underline{89.29}^{\uparrow\ +7.14}$ | $76.34^{\downarrow\ -5.81}$ |
| $\Delta_{\text{LoRA}}^{\text{avg}}$ | | -17.96 | -5.04 | 41.52 | 47.45 |

Table 4: Baseline comparison with other PEFT methods. Performance is reported as effective accuracy—i.e, the geometric mean of the True-Positive and True-Negative rates ($\uparrow$ : improvement relative to LoRA, $\downarrow$ : drop relative to LoRA). Random guessing yields an effective accuracy of 50. Hyperparameters are tuned via sweeps: Prompt Tuning (virtual tokens $\in \{1, 5, 10\}$), LoRA (rank $\in \{16, 32, 64\}$), and MHA-RAG ($H \in \{1, 2, 4, 8\}$). Underlined indicates the best effective accuracy.

**Question Answering in PubMedQA.** Unlike molecular-property-prediction tasks, answering questions in PubMedQA requires access to supporting documents. To ensure comparability, fine-tuning baselines are provided with golden documents as additional context sufficient for answering each question (Joren et al., 2024), presented directly to the foundation model during training and inference. In contrast, MHA-RAG encodes these documents into soft-prompt vectors via its multi-headed attention, rather than appending them directly to the model's input. We also include a comparison with Off-the-Shelf model, where the foundation model is queried without access to these documents, serving as a measure of how much parametric knowledge alone supports question answering. MHA-RAG improves over Off-the-Shelf by an average effective-accuracy gain of 30.91, indicating that its soft prompts effectively capture and represent knowledge from golden documents. Lastly, relative to LoRA, MHA-RAG shows an average effective-accuracy drop of $-5.01$, computed as the geometric mean of drops in PubMedQA. This is expected given that it compresses documents into at most eight soft-prompt tokens. However, inference with MHA-RAG in PubMedQA is computationally cheaper than LoRA, PT and IDPG, because the latter fine-tuning baselines incur inference costs at least as high as RAG (see Figure 2) when documents are pre-pended to the model input.

*Findings.* In domains with limited data, MHA-RAG is an effective method for adapting foundation models; it also efficiently compresses supporting documents into compact soft-prompt tokens.

## 5 CONCLUSION

In this paper, we propose MHA-RAG, an attention-based method inspired by RAG that utilize in-context exemplars to enhance performance. Our approach combines soft-prompt-based domain adaptation with an order-invariant architecture to reduce inference cost and stabilize performance. We also provide the number of heads in the architecture as a tunable parameter to control soft-prompt generation across different experimental setups. With a more efficient, effective, and consistent way of representing in-context exemplars, this work aims to position MHA-RAG as an alternative to RAG for adapting language models to domains. In future, we want to study how well MHA-RAG scales to tasks requiring inference with longer documents and how robust it is to 'lost-in-the-middle' problem (Liu et al., 2023), a common issue in RAG that appears with scaling..

## REPRODUCIBILITY STATEMENT

We provide details of experimental settings, hyperparameters, and datasets in Section 4.1. Proofs of theoretical results are included in Appendix A.1. All source code and data processing scripts will be made publicly available with the camera-ready submission.

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

# A APPENDIX

## A.1 MHA-RAG: IN-CONTEXT EXEMPLAR-ORDER INVARIANCE

In this section, we argue that the soft prompt generated by MHA-RAG is invariant to the order of in-context exemplars/documents.

**Claim 1** (Exemplar-Order Invariance). *Let $q$ be a query vector for a given $x$ and for $k = 1 \ldots K$, let $k_k \in \mathcal{R}^d$ and $v_k \in \mathcal{R}^d$ be keys and value vectors derived from each retrieved example. Define the attention weights as $\alpha_k := \frac{e^{s_k}}{\sum_{j=1}^{K} e^{s_j}}$ where score, $s_k := \frac{q \cdot k_k}{\sqrt{d}}$. The attention output for each head is given by $z = \sum_{k=1}^{K} \alpha_k \, v_k$. Then $z$ is invariant under any permutation of the retrieved examples: permuting the indexing of the pairs $\{(k_k, v_k)\}_{k=1}^{K}$ does not change $z$.*

*Proof.* Let $\pi$ be a permutation of $1 \ldots K$ and consider the permuted sequence $(k_{\pi(1)}, v_{\pi(1)}), \ldots, (k_{\pi(K)}, v_{\pi(K)})$. For the permuted sequence, the scores and weights are

$$s_k' = \frac{q \cdot k_{\pi(k)}}{\sqrt{d}} = s_{\pi(k)} \qquad \alpha_k' = \frac{e^{s_k'}}{\sum_{j=1}^{K} e^{s_j'}} = \frac{e^{s_{\pi(k)}}}{\sum_{j=1}^{K} e^{s_{\pi(j)}}}$$

Because $\{s_{\pi(j)} : j = 1, \ldots, K\}$ is a reordering of $\{s_j : j = 1, \ldots, K\}$, we have $\sum_{j=1}^{K} e^{s_{\pi(j)}} = \sum_{j=1}^{K} e^{s_j}$. Therefore, $\alpha_k' = \alpha_{\pi(k)}$.

Now, the head's resulting output under the permuted attention is $z' = \sum_{k=1}^{K} \alpha_k' \, v_{\pi(k)} = \sum_{k=1}^{K} \alpha_{\pi(k)} \, v_{\pi(k)}$. Without loss of generality, reindex the sum by setting $j = \pi(k)$. Because $\pi$ is a bijection, $k \mapsto j$ permutes the index set $\{1, \ldots, K\}$, resulting in $z' = \sum_{j=1}^{K} \alpha_j \, v_j = z$.

Thus $z$ is unchanged by the permutation $\pi$. The result holds for any permutation, so the MHA's per-head output is order invariant. $\square$

## A.2 BENCHMARKING ON MATH REASONING

In this section, we investigate whether MHA-RAG can improve performance on mathematical reasoning tasks. Specifically, we evaluate on *Inequality Math* [train/dev split: 1252/100], an Olympiad-level benchmark that requires proving bounds preserving inequalities and establishing relations between algebraic expressions (Sheng et al., 2025). For retrieval, we compute dense representations of question–exemplar pairs using *Qwen3-Embedding-8B* (Zhang et al., 2025) and rank candidates by cosine similarity.

We report results primarily on *Llama3.2-3B-Instruct*, since models in the Qwen3 family already achieve high off-the-shelf accuracy (e.g., *Qwen3-4B* at $65\%$), where adding retrieved exemplars led to performance drops. This observation is consistent with Sheng et al. (2025), who found that only certain model families benefit from in-domain exemplars. A plausible explanation is that high-performing models may have already been exposed to mathematical reasoning tasks during pre-training, reducing the marginal utility of retrieval.

| **IneqMath** | Off-The-Shelf | RAG | xRAG | xRAG-K | MHA-RAG |
|---|---|---|---|---|---|
| *Llama3.2-3B-Instruct* | | | | | |
| Bound Accuracy | 10.0 | 10.0 | 4.0 | 8.0 | 14.0 |
| Relation Accuracy | 22.00 | 24.0 | 24.0 | 24.0 | 22.0 |
| Final Accuracy | 16.00 | 17.0 | 14.0 | 16.0 | **18.0** |

Table 5: Baseline comparison on *IneqMath* requiring step-by-step reasoning (with $K = 2$). Accuracy is measured as an exact match of derived numerical values with ground-truth. Hperparameters are tuned via sweeps: MHA-RAG ($H \in \{1, 2, 4\}$). **Bold** indicates the best performance. The model is trained for 1 epoch with a learning rate of $3e - 4$.

From Table 5, we observe that MHA-RAG yields a slight improvement over both off-the-shelf and RAG, whereas training with xRAG and xRAG-K results in a performance drop. A possible explanation is that the *Inequality Math* dataset contains multiple problems that rely on the same theorems or follow similar reasoning steps. If retrieval fails to surface such structurally related exemplars, the model cannot fully benefit from exemplar conditioning. Future work could therefore focus on improving retrieval quality—e.g., by incorporating reasoning-aware similarity metrics—which may allow MHA-RAG to better exploit shared problem structure and yield stronger overall gains.

| Benchmarks | K=5 | | | K=10 | | |
|---|---|---|---|---|---|---|
| | c=0 | c=1 | c=5 | c=0 | c=1 | c=5 |
| *Qwen3-4B* | | | | | | |
| BACE | 76.27 | 75.27 | 79.03 | 52.42 | 76.27 | **79.19** |
| BBBP | 82.49 | 68.56 | **88.14** | 80.43 | 70.01 | 87.5 |
| ClinTox | 96.32 | 76.63 | 94.55 | 89.44 | 70.72 | **99.04** |
| PubMedQA | 76.59 | 70.87 | **83.02** | 66.07 | 72.88 | 82.92 |
| *Llama3.2-3B-Instruct* | | | | | | |
| BACE | 66.16 | **79.96** | 78.82 | 54.73 | 74.23 | 77.75 |
| BBBP | 87.61 | 88.99 | **89.74** | 86.80 | 85.44 | 86.94 |
| ClinTox | 94.44 | 88.97 | **98.40** | 89.33 | 94.49 | 88.97 |
| PubMedQA | 76.34 | 76.56 | 83.17 | 72.50 | 76.45 | **83.37** |

Table 6: Effect of re-inserting top-$c$ exemplars in textual form (up to $c = 5$, given a fixed inference budget) into the in-context prompt while still using top-$K$ for soft-prompt computation in MHA-RAG. The $K = 5$, $c = 0$ column corresponds to the MHA-RAG column of Table 1. **Bold** indicates the best effective accuracy. Underlined indicates the value of $c$ that achieves the best effective accuracy for a given $K$.

To investigate whether performance can be further improved, we augment MHA-RAG's soft prompts with a small number of exemplars directly included in the model's context at inference. Specifically, we add the top-$c$ retrieved exemplars alongside the soft prompts and study the effect across different $K$ values.

As shown in Table 6, effective accuracy improves consistently with increasing $c$, with the best results achieved at $c = 5$ for both $K = 5$ and $K = 10$. The FLOPs analysis in Figure 4 reveals that increasing $K$ from 5 to 10 incurs only a minor cost, while increasing $c$ from 0 to 5 leads to a logarithmic rise in FLOPs. These results suggest that $c$ can serve as a tunable knob for balancing accuracy gains against inference cost, allowing practitioners to adapt MHA-RAG to different computational budgets.

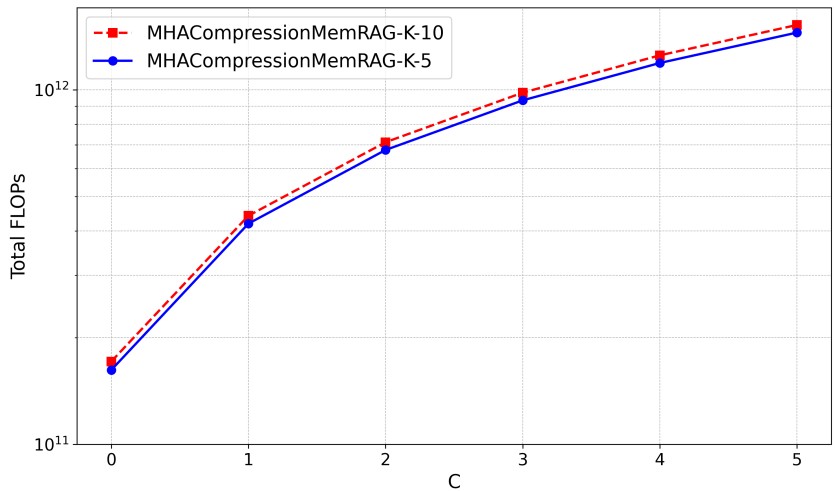

Figure 4: Total FLOPs for inference with encoder ChemBERTa-10M-MTR and foundation model Qwen3-4B using an increasing number of exemplars $c$ in the context, given $K = 5$ and $K = 10$ as input to create a soft prompt.

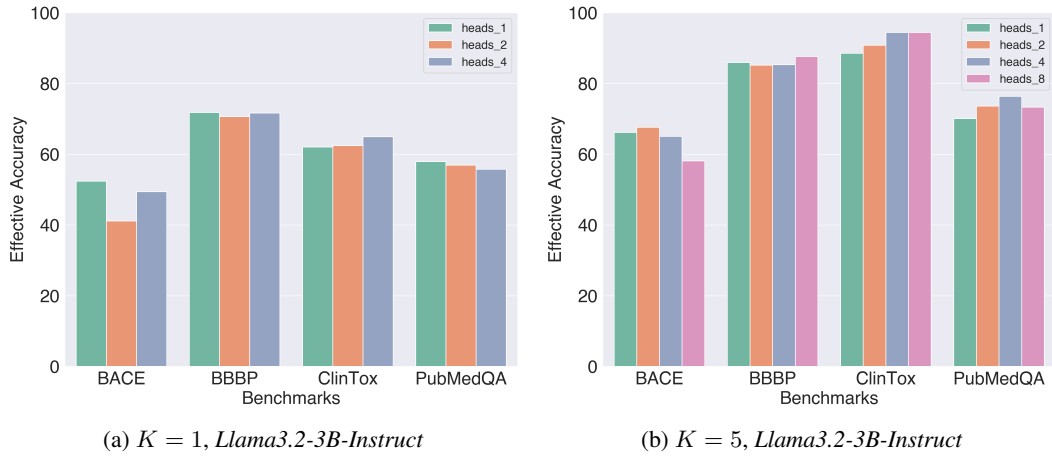

(a) $K = 1$, *Llama3.2-3B-Instruct*          (b) $K = 5$, *Llama3.2-3B-Instruct*

Figure 5: Varying number of heads in MHA-RAG. Performance averaged across 3 seeded runs.

## A.5 TRAINABLE PARAMETERS

| Training Methods | Qwen3-0.6B | Qwen3-4B | Llama3.2-3B-Instruct |
|---|---|---|---|
| *Retrieval-Based Baselines* | | | |
| xRAG | 1.59M | 2.19M | 2.38M |
| xRAG-$K$ | 1.59M | 2.19M | 2.38M |
| MHA-RAG ($m = 1$) | 1.38M | 4.16M | 4.75M |
| MHA-RAG ($m = 2$) | 3.57M | 7.11M | 8.3M |
| MHA-RAG ($m = 4$) | 5.93M | 13.03M | 15.39M |
| *PEFT Baselines* | | | |
| PT ($m = 10$) | 10.24K | 25.6K | 30.72K |
| IDPG ($m = 1$) | 1.59M | 2.19M | 2.38M |
| LoRA ($r = 64$) | 40.37M | 132.12M | 97.26M |

Table 7: Number of trainable parameters when a given foundation model is updated with a specific baseline approach. For retrieval-based baselines, we report overall trainable parameters with *ChemBERTa-2-10M-MTR* as the Encoder.

