# OpenReview forum: "MHA-RAG: Improving Efficiency, Accuracy, and Consistency by Encoding Exemplars as Soft Prompts"
_ICLR.cc/2026/Conference — Submitted to ICLR 2026_

### Official Review · Reviewer_tmhr · 2025-11-01

**Soundness:** 2
**Presentation:** 3
**Contribution:** 2
**Rating:** 4
**Confidence:** 3

**Summary:**

This paper introduces a method that represents exemplars as soft prompts in an invariant manner before feeding them into large language models (LLMs) for question-answer generation. The approach treats the number of attention heads as a tunable hyperparameter to control soft prompt generation across different tasks. Experiments on diverse QA benchmarks show that the proposed MHA-RAG significantly outperforms standard RAG methods and enhances inference efficiency in domain adaptation settings.

**Strengths:**

- The paper presents a strong motivation by clearly identifying the limitations of In-Context Learning (ICL) and the challenges of applying in-context retrieval in Retrieval-Augmented Generation (RAG) for domain adaptation. The proposed approach, which replaces hard token exemplars with soft prompts, proves effective not only in improving task performance (QA accuracy) but also in reducing inference time.
- The paper is generally well-written and well-structured. It includes extensive experiments and ablation studies, offering several insightful empirical findings. These include comprehensive comparisons with traditional prompt tuning, LoRA, and the Instance-Dependent Prompt Generation (IDPG) method.

**Weaknesses:**

- The baseline comparisons are limited, as only xRAG and its variant xRAG-K are included. Given that other related baselines are mentioned, it would strengthen the paper to include comparisons with those methods as well. Furthermore, the work lacks evaluation against other stronger RAG-based approaches (such as adaptive RAG categories) in terms of both task performance and inference cost.
- The proposed method depends on retrieving relevant exemplars from the training set, which may restrict its practicality in domains where no training data is available.
- The experiments are conducted only on relatively small models (up to 4B parameters), leaving the effectiveness of the proposed method on larger-scale LLMs uncertain.

**Questions:**

In Table 1, it is unclear why Qwen3-4B performs significantly worse than Qwen3-0.6B on the ClinTox dataset. Furthermore, after applying MHA-RAG, Qwen3-4B continues to underperform compared to Qwen3-0.6B. Could the authors clarify the reasons behind this unexpected result?

---

> ### Author Response · Authors · 2025-11-21
> **Rebuttal Part 1**
>
> We thank the reviewer for recognizing the motivation and extensive experiments and ablation studies. We provided additional experiments and clarifications to address the reviewer’s questions. We believe that the content from the rebuttal will greatly strengthen our paper.
>
> > **(W1.)”The baseline comparisons are limited, as only xRAG and its variant xRAG-K are included. Given that other related baselines are mentioned, it would strengthen the paper to include comparisons with those methods as well. Furthermore…”**
>
> We agree that comparing more recent benchmarks will be an interesting experiment. However, we want to point out that AttentionRAG and Context Embeddings for Efficient Answer Generation in RAG (COCOM) did not provide any code for running their methods. Therefore, we provide an analysis of their approach in comparison to our approach, based on descriptions from their papers.
>
>  Both AttentionRAG and EXIT operate entirely in the text space, starting from a set of long retrieved documents. Their goal is to prune irrelevant text and retain only the most salient information before feeding the result into a standard RAG pipeline.
>
>  More concretely, AttentionRAG reduces the retrieved context to fewer than ~1000 tokens. It computes attention scores over the prefix and removes text segments with low attention weights; the remaining compressed text is then passed to the foundation model exactly as in classic RAG, where the context is represented as text.
>
> EXIT, by contrast, trains a lightweight classifier that predicts a yes/no decision for each retrieved document. Only documents classified as relevant are included in the foundation-model context for downstream generation. This again follows a standard RAG text-processing pipeline.
>
> In MHA-RAG, we assume a fixed set of retrieved exemplars/documents available to the foundation model prior to entering the MHA-RAG pipeline. Our method directly produces soft prompt tokens in the embedding space, conditioned on the text, rather than manipulating the text itself. Because of this design, the pruning approaches used by AttentionRAG or EXIT are orthogonal and complementary: their attention-based or classifier-based filtering can be applied before MHA-RAG to reduce the retrieved text set, after which MHA-RAG can create a compact set of soft prompt tokens.
>
> Regarding COCOM, it introduces a parameter-efficient fine-tuning scheme that applies LoRA both to the compressor (responsible for document compression) and to the decoder (responsible for generating answers conditioned on the compressed context). In contrast, MHA-RAG learns efficient and expressive representations over exemplars/documents without training or modifying the foundation model itself. Notably, COCOM does not provide training code and only releases pretrained model weights on HuggingFace.
>
> Regarding Adaptive-RAG (cite Adaptive-RAG), it introduces an adaptive QA framework that dynamically selects retrieval strategies based on query complexity predicted by a classifier. In contrast, MHA-RAG focuses on learning compact representations over a fixed set of retrieved exemplars and documents—using standard retrieval functions such as cosine similarity or chemistry-specific similarity metrics—so that the foundation model can consume them efficiently. In principle, Adaptive-RAG could be applied upstream of our method: its adaptive retrieval module could feed exemplars or documents into the MHA-RAG pipeline, where we then learn the multi-head soft-prompt representation. For this reason, we view Adaptive-RAG and similar approaches as orthogonal to our work, rather than competing; our goal is to propose an alternative to RAG itself, not to replace adaptive retrieval mechanisms.
>
> > **(W2.) “The proposed method depends on retrieving relevant exemplars from the training set, which may restrict its practicality in domains where no training data is available.”**
>
> We acknowledge that MHA-RAG requires training data to learn soft-prompt representations of exemplars and documents. This aligns with our goal of providing an alternative to RAG in settings where off-the-shelf foundation models underperform and domain adaptation is needed. In datasets such as BACE, BBBP, and ClinTox, we retrieve question-answer exemplars from domain-specific data; in PubMedQA, we retrieve relevant documents as contextual evidence. Given access to these domain-specific resources, MHA-RAG provides clear advantages over the base model by leveraging exemplars and documents far more efficiently than standard RAG.

---

> ### Author Response · Authors · 2025-11-21
> **Rebuttal Part 2**
>
> > **(W3.) “The experiments are conducted only on relatively small models (up to 4B parameters), leaving the effectiveness of the proposed method on larger-scale LLMs uncertain.”**
>
> In this paper, we considered models from two families - qwen and llama. We include results on Mistral-7B-Instruct-v0.3 (with K=5), a larger LLM from a different family of language models below:
>
> ||RAG|xRAG|xRAG-K|MHA-RAG|
> |:---:|:--:|:--:|:---:|:--:|
> |BACE|77.65|63.92($\downarrow$-13.73)|61.83($\downarrow$-15.82)|76.26($\downarrow$-1.3)|
> |BBBP|66.62|69.46($\uparrow$+2.84)|69.51($\uparrow$+2.89)|87.01($\uparrow$+20.39)|
> |ClinTox|59.79|66.72($\uparrow$+6.93)|69.53($\uparrow$+9.74)|94.42($\uparrow$+34.63)|
> |PubMedQA|79.48|72.83($\downarrow$-6.65)|70.3($\downarrow$-9.18)|73.98($\downarrow$-5.5)|
> |$\Delta^{avg}_{RAG}$||-2.99|-3.61|**10.86**|
>
> As shown in the additional table, MHA-RAG achieves the largest improvement over RAG, with an average Effective Accuracy gain of 10.86. Moreover, MHA-RAG exhibits the smallest performance drop on PubMedQA among all baselines, indicating that it more effectively represents retrieved documents within its soft prompt.
> Overall, these findings demonstrate that the benefits of MHA-RAG extend to other families of models beyond Qwen and LLama.
>
> > **(Q1.) “In Table 1, it is unclear why Qwen3-4B performs significantly worse than Qwen3-0.6B on the ClinTox dataset. Furthermore, after applying MHA-RAG, Qwen3-4B continues to underperform compared to Qwen3-0.6B. Could the authors clarify the reasons behind this unexpected result?”**
>
> As shown in Table 4, both Qwen3-0.6B and Qwen3-4B exhibit low baseline performance on ClinTox (Effective Accuracy = 0.0), with RAG providing clear improvements (Table 1). It is not uncommon for smaller models to sometimes outperform their larger counterparts within the same family on specific tasks, showing that performance on some tasks does not benefit from scaling—this behavior can also be observed in the Qwen3 technical report [1]. We observe a similar pattern on the ClinTox benchmark.
>
> The objective of MHA-RAG is to maintain or improve upon RAG’s performance while addressing its limitations such as high inference cost and exemplar-order variance. MHA-RAG achieves this for both Qwen3-0.6B and Qwen3-4B. Notably, the performance gap between the two models on ClinTox decreases from 9.06 (RAG) to 0.8 (MHA-RAG), indicating that MHA-RAG mitigates much of this discrepancy.
>
> [1] Yang, An, et al. "Qwen3 technical report." arXiv preprint arXiv:2505.09388 (2025).

---

> > ### Comment · Reviewer_tmhr · 2025-11-26
> > **Response to Authors**
> >
> > >(W1)  In principle, Adaptive-RAG could be applied upstream of our method: its adaptive retrieval module could feed exemplars or documents into the MHA-RAG pipeline, where we then learn the multi-head soft-prompt representation. For this reason, we view Adaptive-RAG and similar approaches as orthogonal to our work, rather than competing; our goal is to propose an alternative to RAG itself, not to replace adaptive retrieval mechanisms.
> >
> > I agree that adaptive-based RAG methods are orthogonal and can be applied upstream of the proposed approach. However, these methods also aim to improve the overall efficiency and effectiveness of RAG systems. Providing a comparison between adaptive-based and compression-based methods and highlighting their respective advantages and limitations would be valuable to the community.
> >
> > I appreciate the authors’ detailed response. However, since my concerns remain (W1 and W2), I will retain my current scores.

---

> ### Author Response · Authors · 2025-11-26
> **Follow-up**
>
> **Regarding W1:**
>
> We agree that adaptive-based RAG methods also aim to improve the efficiency and effectiveness of RAG systems. To further strengthen our comparison, we additionally evaluated MHA-RAG against EXIT (an adaptive compression-based method) on PubMedQA using Mistral-7B-Instruct-v0.3 with (K=5). We report Effective Accuracy and inference time below:
>
> |          Metric          |   RAG  |              EXIT              |            MHA-RAG            |
> | :----------------------: | :----: | :----------------------------: | :---------------------------: |
> |    Effective Accuracy    |  79.48 |   67.31 ($\downarrow$ -12.17)   |   73.98 ($\downarrow$ -5.50)   |
> | Avg. Inference Time (ms) | 1531.6 | 1321.3 ($\uparrow$ +13.7%) | 481.5 ($\uparrow$ +68.6%) |
>
> We observe that compressing retrieved context using MHA-RAG (in the input embedding space) is more effective than compressing via EXIT (in the text space). While EXIT saves inference time by 13.7%, this comes with a larger drop in Effective Accuracy relative to RAG. In contrast, MHA-RAG reduces inference time by +68.6% while incurring a smaller drop in accuracy. This establishes MHA-RAG as a more effective compression strategy compared to EXIT.
>
> Additionally, we highlight two advantages of MHA-RAG beyond performance-efficiency trade-offs. Adaptive compression methods such as EXIT are naturally suited to document-based retrieval, but their deployment is non-trivial when the retrieved context consists of in-context exemplars (as in BBBP, BACE, and ClinTox). MHA-RAG supports both document-based and exemplar-based settings. Further, compression in text space inherits RAG’s order variance, whereas MHA-RAG is inherently order-invariant by design.
>
> **Regarding W2:**
>
> We acknowledge that MHA-RAG requires some training, whereas classical RAG can operate without any. Our aim, in this paper, is not to replace RAG in settings with zero available data, but to provide a more efficient and effective alternative in the frequent scenario where limited domain-specific data exists but full model fine-tuning is impractical. In this regime, with a modest amount of training, MHA-RAG delivers superior performance–efficiency trade-offs compared to both adaptive and compression-based baselines, making it a practical domain adaptation strategy
>
> We hope that, together, these results and observations address the concerns in W1 and W2. We will include the EXIT baseline comparisons in the final version of the manuscript to strengthen the contribution.

---

### Official Review · Reviewer_VyAD · 2025-11-01

**Soundness:** 2
**Presentation:** 3
**Contribution:** 2
**Rating:** 2
**Confidence:** 4

**Summary:**

This paper introduces Multi-Head Attention Retrieval-Augmented Generation (MHA-RAG), a framework that represents retrieved in-context exemplars as soft prompts using multi-head attention mechanisms. The key innovation is using the number of attention heads as a tunable hyperparameter to control soft-prompt generation, claiming to achieve order-invariance through scaled dot-product attention. The authors evaluate MHA-RAG on molecular property prediction tasks (BACE, BBBP, ClinTox) and biomedical QA (PubMedQA) using small language models (Qwen3-0.6B/4B, Llama3.2-3B-Instruct). Results show an average 20-point performance gain over standard RAG while reducing inference costs by 10× in terms of GFLOPs.

**Strengths:**

1. The theoretical guarantee of order-invariance through scaled dot-product attention is sound and formally proven. This addresses a real problem in RAG systems where exemplar order can affect performance (Table 2 shows clear variance in baselines).

2. MHA-RAG shows consistent performance gains over RAG and xRAG variants across the molecular property prediction tasks and PubMedQA (Table 1), with an average improvement of 19.66 points.

3. The paper includes useful ablations on number of heads (Figures 3, 5) and value of K (Table 3), showing that performance generally improves with more heads when sufficient context is available.

4. Appendix A.3 (Table 6) provides interesting analysis showing that adding back a small number of text exemplars (c=5) alongside soft prompts can further improve performance, suggesting the soft prompts and text are complementary.

**Weaknesses:**

1. Missing Baselines: The paper does not compare against crucial recent methods:
* AttentionRAG: Attention-Guided Context Pruning in Retrieval-Augmented Generation, which uses model attention to prune retrievals in a training-free manner.
* EXIT: Context-Aware Extractive Compression for Enhancing Retrieval-Augmented Generation, a Context-aware extractive compression for RAG that preserves contextual dependencies. Reported to outperform both compressed and uncompressed baselines.
* Context Embeddings for Efficient Answer Generation in RAG, which compresses contexts into embeddings for RAG, handles multiple documents, offers tunable compression rates. This is a direct competitor that should be compared.

Without these comparisons, we can't assess whether MHA-RAG represents an improvement over current SOTA or is merely competitive with outdated baselines (xRAG from NeurIPS 2024).


2. The efficiency claims are undermined by missing measurements:
* Only GFLOPs are reported, but additional encoding overhead from sentence embedding model is not counted. Meanwhile, the attention computation over K exemplars with H heads has its own cost. Moreover, there is no comparison of actual inference time on identical hardware.
* The paper doesn't report 1)Peak memory consumption during inference, 2)M emory required to store embedding model + foundation model, and 3) Memory footprint compared to RAG with FlashAttention, which is a common optimization used in applications

**Questions:**

* Can you provide more comparisons with the recent baselines mentioned in the weakness?
* Modern RAG systems use 7B+ models. Can you provide results on Llama-3-8B, Mistral-7B, or similar scales to test if the method scale?
* Can you provide more details around the efficiency, including 1) wall-clock latency measurements on identical hardware, 2) peak memory consumption during training and inference, 3) complete FLOPs including encoding overhead, and 4) comparison with FlashAttention-enabled RAG?

ps:
* Table 4 shows LoRA achieving 0.0 effective accuracy on 9 out of 12 settings. This is unreasonable for a well-established method. Can you provide learning curves, loss plots, etc. to explain this?
* Table 3 shows MHA-RAG performance degrades at K=10 compared to K=5, does it mean the proposed methods may not scale to more examples?

---

> ### Author Response · Authors · 2025-11-21
> **Rebuttal Part 1**
>
> We thank the reviewer for recognizing the strengths of our paper, including its theoretical guarantees, consistent performance gains, ablation studies, and in-depth analysis. We have provided additional experiments and clarifications to address the reviewer’s concerns. We believe that the content from the rebuttal will greatly strengthen our paper.
>
> > **(W1.) “The efficiency claims are undermined by missing measurements: Only GFLOPs are reported, but additional encoding overhead from sentence embedding model is not counted. Meanwhile, the attention computation over K exemplars with H heads has its own cost.”**
>
> GLOPs computation shown in Figure 2 for xRAG and MHA-RAG already includes the corresponding soft-prompt encoding overheads from the sentence embedding model and attention computation from exemplars with H heads in MHA-RAG. Figure 2 indicates end-to-end GFLOPS computed for all methods.
>
> > **(W2.) “Moreover, there is no comparison of actual inference time on identical hardware. The paper doesn't report 1) Peak memory consumption during inference”**
>
> To address it, we report the average answer-generation time (ms) and peak GPU memory consumption (GB) on identical hardware in the table below:
>
> |Qwen3-4B on BACE (K=5)|RAG|MHA-RAG (m=1->2->4->8)|
> |:---:|:--:|:--:|
> |Peak Memory (MB)|8049.16|7836.33→7841.32→7854.07→7880.41|
> |Average Inference Time (ms)|524.3|318.7→ 303.9→ 314.7→ 320.2|
>
> These measurements were obtained using the Qwen3-4B foundation model on the BACE benchmark with K=5 for all retrieval-based baselines. Across this setting, we observe that representing retrieved exemplars via MHA-RAG yields both lower inference latency and reduced GPU memory usage compared to RAG, and this trend holds across different numbers of attention heads.
>
> > **(W3.) “Missing Measurements:  2) Memory required to store embedding model + foundation model, and 3) Memory footprint compared to RAG with FlashAttention, which is a common optimization used in applications.”**
>
> Regarding storage requirements, we note that—unlike COCOM [1]—MHA-RAG does not store precomputed embedding indexes for the training database because they are computed on the fly for each inference-time question. This design avoids additional memory overhead associated with storing vectors.
>
> Regarding FlashAttention, our experiments are conducted on V100 GPUs, which unfortunately do not support FlashAttention. FlashAttention primarily reduces memory usage and improves speed, but it does not change FLOPs. For fairness, MHA-RAG uses the same foundation models as RAG. The encoders we use (Qwen3-0.6B and ChemBERTa) could, in principle, also benefit from FlashAttention. Therefore, even if FlashAttention were enabled, we would not expect a significant difference in memory usage between RAG and MHA-RAG.
>
> [1] Rau, David, et al. "Context embeddings for efficient answer generation in rag." arXiv preprint arXiv:2407.09252 (2024).

---

> > ### Author Response · Authors · 2025-11-21
> > **Rebuttal Part 2**
> >
> > > **(Q1.) “Missing Baselines: The paper does not compare against crucial recent methods:...”**
> >
> > We agree that comparing more recent benchmarks will be an interesting experiment. However, we want to point out that AttentionRAG and Context Embeddings for Efficient Answer Generation in RAG (COCOM) did not provide any code for running their methods. Therefore, we provide an analysis of their approach in comparison to our approach, based on descriptions from their papers.
> >
> > Both AttentionRAG and EXIT operate entirely in the text space, starting from a set of long retrieved documents. Their goal is to prune irrelevant text and retain only the most salient information before feeding the result into a standard RAG pipeline.
> > More concretely, AttentionRAG reduces the retrieved context to fewer than ~1000 tokens. It computes attention scores over the prefix and removes text segments with low attention weights; the remaining compressed text is then passed to the foundation model exactly as in classic RAG, where the context is represented as text.
> >
> > EXIT, by contrast, trains a lightweight classifier that predicts a yes/no decision for each retrieved document. Only documents classified as relevant are included in the foundation-model context for downstream generation. This again follows a standard RAG text-processing pipeline.
> >
> > In MHA-RAG, we assume a fixed set of retrieved exemplars/documents available to the foundation model prior to entering the MHA-RAG pipeline. Our method directly produces soft prompt tokens in the embedding space, conditioned on the text, rather than manipulating the text itself. Because of this design, the pruning approaches used by AttentionRAG or EXIT are orthogonal and complementary: their attention-based or classifier-based filtering can be applied before MHA-RAG to reduce the retrieved text set, after which MHA-RAG can create a compact set of soft prompt tokens.
> >
> > Regarding COCOM, it introduces a pretraining stage and a parameter-efficient fine-tuning stage that applies LoRA to both the compressor (responsible for document compression) and the decoder (responsible for generating answers conditioned on the compressed context). In contrast, MHA-RAG learns efficient and expressive representations over exemplars/documents without pre-training or fine-tuning on the foundation model itself. It would be interesting to compare against COCOM; however, they only released the pretrained model weights on HuggingFace. They do not provide the data splits they used or any fine-tuning code, which makes it hard to perform a fair and reproducible comparison with our methods. We’ll do our best to see what we can realistically try with this method.
> >
> > > **(Q2.) “Modern RAG systems use 7B+ models. Can you provide results on Llama-3-8B, Mistral-7B, or similar scales to test if the method scale?”**
> >
> > In this paper, we considered models from two families - qwen and llama. We include results on Mistral-7B-Instruct-v0.3 (with K=5), a larger LLM from a different family of language models below:
> >
> > ||RAG|xRAG|xRAG-K|MHA-RAG|
> > |:---:|:--:|:--:|:---:|:--:|
> > |BACE|77.65|63.92($\downarrow$-13.73)|61.83($\downarrow$-15.82)|76.26($\downarrow$-1.3)|
> > |BBBP|66.62|69.46($\uparrow$+2.84)|69.51($\uparrow$+2.89)|87.01($\uparrow$+20.39)|
> > |ClinTox|59.79|66.72($\uparrow$+6.93)|69.53($\uparrow$+9.74)|94.42($\uparrow$+34.63)|
> > |PubMedQA|79.48|72.83($\downarrow$-6.65)|70.3($\downarrow$-9.18)|73.98($\downarrow$-5.5)|
> > |$\Delta^{avg}_{RAG}$||-2.99|-3.61|**10.86**|
> >
> > As shown in the additional table, MHA-RAG achieves the largest improvement over RAG, with an average Effective Accuracy gain of 10.86. Moreover, MHA-RAG exhibits the smallest performance drop on PubMedQA among all baselines, indicating that it more effectively represents retrieved documents within its soft prompt.
> > Overall, these findings demonstrate that the benefits of MHA-RAG extend to other families of models (and larger models) beyond Qwen and LLama.

---

> > > ### Author Response · Authors · 2025-11-21
> > > **Rebuttal Part 3**
> > >
> > > > **(Q3.) “Table 4 shows LoRA achieving 0.0 effective accuracy on 9 out of 12 settings. This is unreasonable for a well-established method. Can you provide learning curves, loss plots, etc. to explain this?”**
> > >
> > > |step|87|174|261|348|435|
> > > |:---:|:--:|:--:|:---:|:--:|:--:|
> > > |Train Loss|1.9904|0.1176|0.0976|0.0849|0.0699|
> > > |Eval Loss|0.1564|0.0828|0.0827|0.0885|0.0768|
> > >
> > > We include the training and evaluation loss across training steps for Qwen3-4B model on ClinTox dataset under LoRA fine-tuning, which ultimately yields 0.0 effective accuracy. For the yes/no classification tasks, we report the geometric mean (g-mean) of the true-positive and true-negative rates, since both classes are equally important. A model that predicts all yes or all no will have a g-mean of 0.0, which is exactly what appears in Table 4 for many of the LoRA runs.
> > >
> > > The training and evaluation loss shown above confirm this behavior: although the loss decreases on both splits—indicating the model is fitting the training data—the resulting predictions provide no useful discrimination between classes. This is consistent with the highly imbalanced label distributions in BBBP and ClinTox, where LoRA tends to overfit while failing to generalize.
> > >
> > > > **(Q4.) “Table 3 shows MHA-RAG performance degrades at K=10 compared to K=5, does it mean the proposed methods may not scale to more examples?”**
> > >
> > > Our goal in Table 3 is not to argue that “more exemplars is always better,” but to study how much retrieved information is actually needed before additional context becomes counter-productive. In Section 4.3, we explicitly frame this issue as context saturation: as K increases from 1 → 5, both RAG and MHA-RAG consistently improve, and MHA-RAG at K=5 already outperforms RAG at both K=5 and K=10 while using many fewer FLOPs. Beyond that point, pushing K to 10 mostly adds lower-quality or redundant exemplars, which is known to introduce noise and can degrade effective accuracy. This is a property of the data and retrieval quality, not a fundamental limitation of MHA-RAG.
> > >
> > > In practice, we treat K as a standard hyperparameter and select the smallest K that achieves peak performance for a given domain. Under that view, Table 3 actually demonstrates that MHA-RAG scales up to the point where additional retrieved examples cease to be informative, and that it reaches this “sufficient context” regime earlier and more compute-efficiently than RAG. Exploring regimes with more training data and higher-precision retrieval to push that saturation point further out is an interesting but orthogonal direction to the current work.

---

> ### Author Response · Authors · 2025-11-26
> **Follow Up for Q1**
>
> **Quantitative Comparison with Adaptive-RAG-based Baseline**
>
> While the code for AttentionRAG is not available, to further strengthen our comparison, we additionally evaluated MHA-RAG against EXIT (an adaptive compression-based method) on PubMedQA using Mistral-7B-Instruct-v0.3 with (K=5). We report the Effective Accuracy and inference time below:
>
> |          Metric          |   RAG  |              EXIT              |            MHA-RAG            |
> | :----------------------: | :----: | :----------------------------: | :---------------------------: |
> |    Effective Accuracy    |  79.48 |   67.31 ($\downarrow$ -12.17)   |   73.98 ($\downarrow$ -5.50)   |
> | Avg. Inference Time (ms) | 1531.6 | 1321.3 ($\uparrow$ +13.7%) | 481.5 ($\uparrow$ +68.6%) |
>
> We observe that compressing retrieved context using MHA-RAG (in the input embedding space) is more effective than compressing via EXIT (in the text space). While EXIT saves inference time by 13.7%, this comes with a larger drop in Effective Accuracy relative to RAG. In contrast, MHA-RAG reduces inference time by +68.6% while incurring a smaller drop in accuracy. This establishes MHA-RAG as a more effective compression strategy compared to EXIT.
>
> Additionally, we highlight two advantages of MHA-RAG beyond performance-efficiency trade-offs. Adaptive compression methods such as EXIT are naturally suited to document-based retrieval, but their deployment is non-trivial when the retrieved context consists of in-context exemplars (as in BBBP, BACE, and ClinTox). MHA-RAG supports both document-based and exemplar-based settings. Further, compression in text space inherits RAG’s order variance, whereas MHA-RAG is inherently order-invariant by design.

---

### Official Review · Reviewer_qhuv · 2025-11-01

**Soundness:** 2
**Presentation:** 3
**Contribution:** 2
**Rating:** 2
**Confidence:** 5

**Summary:**

In this paper, the authors propose multi-head attention retrieval-augmented generation to adapt foundation models to new domains with limited data, thereby addressing 3 challenges for conventional RAG, including high inference costs from long textual contexts, suboptimal performance on out-of-distribution data, and sensitivity to the order of retrieved exemplars. Instead of appending exemplars as text, MHA-RAG encodes them into compact soft prompts using multi-head scaled dot-product attention and ensures order invariance and achieves high compression ratios.

The experiments are conducted on low-data molecular property prediction tasks and  biomedical QA. The results show the improvements in effective accuracy and FLAPs reductions. The authors also conclude by positioning the proposed method as a cost-effective RAG alternative with future work on scaling to longer documents.

**Strengths:**

* Strong empirical gains in accuracy while reducing educing inference costs
* Multi-head attention addresses the performance fluctuation issue of RAG with true order invariance
* The hyperparameter offers some flexibility to tune trade-offs between effectiveness and efficiency across models and domains

**Weaknesses:**

* Limited scope of benchmarks that only focus on specific domains in the experiments.
* Lack of robustness analysis for noisy and adversarial exemplars
* Missing discussion on training efficiency

**Questions:**

* How does MHA-RAG perform on open-ended text generation tasks, such as summarization or creative writing?
* I wonder if the authors can provide some specific examples of failure cases where MHA-RAG produces incorrect outputs.
* The retrieval function could matter. I wonder if the authors have studied on the choice of this.
* What results does MHA-RAG achieve when using foundation models from other families, like GPT or Mistral?

---

> ### Author Response · Authors · 2025-11-21
> **Rebuttal Part 1**
>
> We thank the reviewer for the comments and the opportunity to clarify some of the ideas in our paper. We have provided additional experiments and clarifications to address the reviewer’s concerns. We believe that the content from the rebuttal will greatly strengthen our paper.
>
> >**(W1.) “Missing discussion on training efficiency”**
>
> In Table 7 from Appendix A.5, we provide the number of trainable parameters for each baseline, where we can observe that MHA-RAG with the number of heads as a tunable parameter offers flexibility in controlling the performance with parameter-efficiency. It is more parameter-efficient than LoRA and similar to other retrieval baselines; it requires a few million trainable parameters but provides a competitive edge over them (Table 1).
>
> >**(Q1.) “Limited scope of benchmarks that only focus on specific domains in the experiments. How does MHA-RAG perform on open-ended text generation tasks, such as summarization or creative writing?”**
>
> In this paper, we are interested in finding effective and efficient representations over exemplars or documents in which these exemplars and documents help answer questions. Papers have shown that including retrieved exemplars or documents in the context do help but rather some special mechanisms such as chain-of-thought [1] or graph structured data [2] need to be used with open-ended text generation tasks, summarisation task or creative writing which are out of the scope of this paper.
> We include results on the open-domain generation benchmark of HotPotQA:
>
> |Qwen3-0.6B|Base|RAG|xRAG|xRAG-K|MHA-RAG (# heads)|
> |:---:|:--:|:--:|:---:|:--:|:--:|
> |Exact Match|0.0|**16.21**|10.49|8.32|**11.8** (8)|
>
> |Llama-3.2-3B|Base|RAG|xRAG|xRAG-K|MHA-RAG (# heads)|
> |:---:|:--:|:--:|:---:|:--:|:--:|
> |Exact Match|12.5|**27.7**|22.01|21.99|**22.71** (8)|
>
> Our results show that MHA-RAG substantially outperforms the off-the-shelf base LLM, confirming that adding retrieved context—even in compressed soft-prompt form—meaningfully improves open-domain QA performance. Compared to compression-based methods (xRAG and xRAG-K), MHA-RAG also achieves higher accuracy, demonstrating that the multi-head attention mechanism is more effective at extracting salient information from long documents.
>
> Finally, while MHA-RAG achieves within 5% accuracy drop compared to full RAG, it uses only 8 soft-prompt tokens to represent documents that RAG expands to roughly 650 tokens on average. This compression reduces inference cost by about 10X (Figure 2), yet preserves most of RAG’s performance—highlighting that MHA-RAG generalizes to open-domain QA while offering a significantly more efficient alternative.
>
> [1] Joren, Hailey, et al. "Sufficient context: A new lens on retrieval augmented generation systems." arXiv preprint arXiv:2411.06037 (2024).
> [2] Edge, Darren, et al. "From local to global: A graph rag approach to query-focused summarization." arXiv preprint arXiv:2404.16130 (2024).
>
> > **(Q2.) “Lack of robustness analysis for noisy and adversarial exemplars. The retrieval function could matter. I wonder if the authors have studied on the choice of this.”**
>
> To assess robustness to noisy exemplars, we conducted an experiment where exemplars were randomly sampled from the database instead of selecting the top-5 most relevant ones (the number of attention heads for MHA-RAG is indicated in parentheses) below:
>
> |Qwen3-4B|RAG (Top-5)|RAG (Random-5)|MHA-RAG (Top-5)|MHA-RAG (Random-5)|
> |:---:|:--:|:--:|:---:|:--:|
> |BACE|75.87|54.37|**76.27**|58.31|
> |BBBP|69.34|59.36|**82.49**|68.26|
> |ClinTox|44.24|0.0|**96.32**|83.22|
>
> |Llama3.2-3B|RAG (Top-5)|RAG (Random-5)|MHA-RAG (Top-5)|MHA-RAG (Random-5)|
> |:---:|:--:|:--:|:---:|:--:|
> |BACE|49.89|52.45|**67.62**|52.49|
> |BBBP|46.15|50.74|**87.61**|64.55|
> |ClinTox|37.90|39.40|**94.44**|88.97|
>
> We observed that both RAG and MHA-RAG experience a performance drop under this setting, particularly with Qwen3-4B; however, MHA-RAG consistently outperforms RAG. This indicates that (1) exemplar relevance plays a critical role in effective question answering, and (2) MHA-RAG is better at representing and leveraging even randomly sampled (i.e., noisy) exemplars than RAG.
> Interestingly, for Llama3.2-3B-Instruct, RAG shows a slight improvement when using random exemplars, while MHA-RAG exhibits a drop. Nevertheless, MHA-RAG continues to outperform RAG under both random and relevant exemplar settings, further supporting its superior representational capability.

---

> ### Author Response · Authors · 2025-11-21
> **Rebuttal Part 2**
>
> >**(Q3.) "I wonder if the authors can provide some specific examples of failure cases where MHA-RAG produces incorrect outputs."**
>
> We share two representative failure cases for MHA-RAG and RAG from HotpotQA below:
>
> **First question**: The director of the romantic comedy “Big Stone Gap” is based in what New York city?
>
> Context: … Big Stone Gap is a 2014 American drama romantic comedy film written and…the story is set in the actual Virginia town of Big Stone Gap… (Answer “Greenwich Village” does not appear in context)
>
> Correct Answer: Greenwich Village, New York City
>
> RAG answer: Adriana Trigiani
>
> MHA-RAG answer: Manhattan
>
> In this example, both RAG and MHA-RAG fail to produce a fully correct answer. RAG’s prediction is entirely incorrect and reflects a misunderstanding of the question, which asks for a geographic location. MHA-RAG, while still wrong, provides a partially correct answer by predicting Manhattan where Greenwich Village resides, but is penalised by the Exact Match-like metric.
>
> **Second question**: When did the English local newspaper, featuring the sculpture and war memorial in the Forbury gardens, change names?
>
> Context: The Reading Post (until 2009, the Reading Evening Post), was an English localnewspaper covering Reading…The memorial is in the memory of the soldiers and officers of the Indian Army who were killed during the 1999 conflict…
>
> Correct Answer: 2009
>
> RAG: 2009
>
> MHA-RAG: 1999
>
> In this example, RAG answers correctly by identifying the year 2009. MHA-RAG also understands that the question requires a year, but it selects the wrong one because it compresses the entire context into only a few soft-prompt tokens. This extremely compact representation can occasionally lose fine-grained details, leading the model to pick an incorrect year despite understanding the question type.
>
> > **(Q4.) “What results does MHA-RAG achieve when using foundation models from other families, like GPT or Mistral?”**
>
> In this paper, we considered models from two families - qwen and llama. We want to point that GPT family of models are closed source and thus can’t be used in our framework because MHA-RAG requires back propagating gradients through the foundation model. We include results on Mistral-7B-Instruct-v0.3 (with K=5), a larger LLM from a different family of language models below:
>
> ||RAG|xRAG|xRAG-K|MHA-RAG|
> |:---:|:--:|:--:|:---:|:--:|
> |BACE|77.65|63.92($\downarrow$-13.73)|61.83($\downarrow$-15.82)|76.26($\downarrow$-1.3)|
> |BBBP|66.62|69.46($\uparrow$+2.84)|69.51($\uparrow$+2.89)|87.01($\uparrow$+20.39)|
> |ClinTox|59.79|66.72($\uparrow$+6.93)|69.53($\uparrow$+9.74)|94.42($\uparrow$+34.63)|
> |PubMedQA|79.48|72.83($\downarrow$-6.65)|70.3($\downarrow$-9.18)|73.98($\downarrow$-5.5)|
> |$\Delta^{avg}_{RAG}$||-2.99|-3.61|**10.86**|
>
> As shown in the additional table, MHA-RAG achieves the largest improvement over RAG, with an average Effective Accuracy gain of 10.86. Moreover, MHA-RAG exhibits the smallest performance drop on PubMedQA among all baselines, indicating that it more effectively represents retrieved documents within its soft prompt.
> Overall, these findings demonstrate that the benefits of MHA-RAG extend to other families of models beyond Qwen and LLama.

---

### Official Review · Reviewer_iMqW · 2025-11-03

**Soundness:** 3
**Presentation:** 3
**Contribution:** 2
**Rating:** 4
**Confidence:** 4

**Summary:**

This paper introduces MHA-RAG, a model that encodes each retrieved document as distinct key and value embeddings. A multi-head attention (MHA) layer is then applied to the query, key, and value embeddings. Each head within the MHA layer learns a unique embedding. These head embeddings are subsequently concatenated to form a fixed-size embedding matrix for any given set of k documents.
The experimental evaluation of MHA-RAG was conducted in domains where the base Large Language Model (LLM) exhibits suboptimal performance, thereby necessitating the integration of retrieved documents. The experimental results align with the theoretical analysis, confirming MHA-RAG's order invariance. Through careful selection of hyperparameters, such as K and H, MHA-RAG demonstrates superior performance compared to baseline models in most evaluated scenarios.

**Strengths:**

1. The proposed method is clearly presented and easy to follow.
2. Extensive experiments were conducted to prove the effectiveness of the MHA-RAG, and the effects of different hyperparameters were also studied.
3. MHA-RAG is justified as being order-invariant, a characteristic that cannot be guaranteed by the baseline methods.

**Weaknesses:**

1. Limited Generalization: All datasets utilize pre-defined labels (e.g., "Yes" or "No"), which raises concerns regarding the model's ability to generalize to open-domain Question Answering (QA) tasks.
2. Inconsistent Performance: MHA-RAG fails to consistently outperform the basic RAG baseline (as shown in Table 1), and the reasons for this inconsistency are not explained.
3. Incomplete Comparison (Table 1): Table 1 exclusively presents performances with K=5. This limited scope raises concerns about a fair comparison, as other baseline methods might perform better with a different value of K.
4. Missing Hyperparameter Details (Table 3): While Table 3 studies the effect of varying K, the corresponding value of H used in these experiments is not provided.

**Questions:**

1. Section **Performance Metrics** (line 273): what is the range of o_{i}?
2. Figure 1 shows that MHA-RAG freeze foundation model and only train the encoder the multi-head attention. Can you clarify in *Training Details* (line 309), what does the `Models` refer to?
3. What is the metric for evaluating PubMedQA dataset in **Performance Metrics**?
4. What is the number of attention heads in Table 3?

---

> ### Author Response · Authors · 2025-11-21
> **Rebuttal Part 1**
>
> We thank the reviewer for recognizing our contributions and for the opportunity to provide additional experiments and clarifications about MHA-RAG’s generalization and performance analysis. The content from the rebuttal will greatly strengthen our paper.
> > **(W1.) “Limited Generalization: All datasets utilize pre-defined labels (e.g., "Yes" or "No"), which raises concerns regarding the model's ability to generalize to open-domain Question Answering (QA) tasks.”**
>
> We address this concern directly by evaluating MHA-RAG on another open-domain QA benchmark called HotpotQA [1] beyond PubMedQA in the main paper and Math Reasoning (shown in Appendix A.2) from the paper. HotpotQA requires multi-hop reasoning over long, unstructured supporting documents and generates natural language answers, rather than simply “Yes” or “No”. In this setting, MHA-RAG encodes the retrieved documents into only a small number of soft-prompt vectors, whereas baselines such as RAG operate on hundreds of raw tokens. The results are shown below:
>
>
> |Qwen3-0.6B|Base|RAG|xRAG|xRAG-K|MHA-RAG (# heads)|
> |:---:|:--:|:--:|:---:|:--:|:--:|
> |Exact Match|0.0|**16.21**|10.49|8.32|**11.8** (8)|
>
> |Llama-3.2-3B|Base|RAG|xRAG|xRAG-K|MHA-RAG (# heads)|
> |:---:|:--:|:--:|:---:|:--:|:--:|
> |Exact Match|12.5|**27.7**|22.01|21.99|**22.71** (8)|
>
> Our results show that MHA-RAG substantially outperforms the off-the-shelf base LLM, confirming that adding retrieved context—even in compressed soft-prompt form—meaningfully improves open-domain QA performance. Compared to compression-based methods (xRAG and xRAG-K), MHA-RAG also achieves higher accuracy, demonstrating that the multi-head attention mechanism is more effective at extracting salient information from long documents.
> Finally, while MHA-RAG achieves within 5% accuracy drop compared to full RAG, it uses only 8 soft-prompt tokens to represent documents that RAG expands to roughly 650 tokens on average. This compression reduces inference cost by about 10X (as shown in Figure 2), yet preserves most of RAG’s performance—highlighting that MHA-RAG generalizes to open-domain QA while offering a significantly more efficient alternative.
>
> [1] Yang, Zhilin, et al. "HotpotQA: A dataset for diverse, explainable multi-hop question answering." Proceedings of the 2018 conference on empirical methods in natural language processing. 2018.
>
> > **(W2.) "Inconsistent Performance: MHA-RAG fails to consistently outperform the basic RAG baseline (as shown in Table 1), and the reasons for this inconsistency are not explained"**
>
> As shown in Table 1, MHA-RAG outperforms the standard RAG baseline in 10 out of 12 cases, with only minor drops of 0.93 and 2.67 in the remaining two. These decreases are notably smaller than those observed for other baselines (xRAG and xRAG-K). However, even if MHA-RAG were consistently outperformed by RAG, MHA-RAG (as well as xRAG and xRAG-K) still have the benefit of substantially reducing inference costs (Figure 2), and would have significant value, even in that case.
>
> > **(W3.) "Incomplete Comparison (Table 1): Table 1 exclusively presents performances with K=5. This limited scope raises concerns about a fair comparison, as other baseline methods might perform better with a different value of K"**
>
> We report the results for xRAG and xRAG-K with K=1,5,10 below:
>
> |Qwen3-4B|xRAG (K=1)|xRAG (K=5)|xRAG (K=10)|xRAG-K (K=1)|xRAG-K (K=5)|xRAG-K (K=10)|MHA-RAG (K=5)|
> |:---:|:--:|:--:|:---:|:--:|:--:|:--:|:--:|
> |BACE|61.35|59.07|34.03|37.57|55.46|56.87|**76.27**|
> |BBBP|73.51|64.44|70.43|68.26|81.40|67.46|**82.49**|
> |ClinTox|47.64|40.19|65.16|46.71|57.07|59.55|**96.32**|
> |PubMedQA|55.28|71.16|74.52|55.05|70.36|70.56|**76.59**|
>
> |Qwen3-4B|xRAG (K=1)|xRAG (K=5)|xRAG (K=10)|xRAG-K (K=1)|xRAG-K (K=5)|xRAG-K (K=10)|MHA-RAG (K=5)|
> |:---:|:--:|:--:|:---:|:--:|:--:|:--:|:--:|
> |BACE|39.77|54.85|16.29|22.68|64.04|18.97|**67.62**|
> |BBBP|69.80|72.87|15.81|71.61|84.09|70.49|**87.61**|
> |ClinTox|67.35|62.88|72.54|60.08|86.89|92.80|**94.44**|
> |PubMedQA|55.87|75.08|73.21|55.66|75.71|71.84|**76.34**|
>
> As noted, in many cases we can improve the performance of baseline methods such as xRAG, and xRAG-K by considering K as an hyper-parameter. However, MHA-RAG at ( K = 5 ) still consistently outperforms xRAG and xRAG-K at ( K = 1, 5, 10 ). This reinforces our finding (lines 376–377) that MHA-RAG more effectively extracts and represents information necessary for accurate question answering. We will append these findings to those of Table 3 in the paper.

---

> > ### Author Response · Authors · 2025-11-21
> > **Rebuttal Part 2**
> >
> > > **(Q1.) “Section Performance Metrics (line 273): what is the range of o_{i}?”**
> >
> > $o_{i}$ denotes the score of a method on a specific task (i.e., a given foundation model evaluated on a particular benchmark). Since we use Effective Accuracy as the evaluation metric,  $o_i$  ranges from 0 to 100. For example, $o_{Qwen3\text{-}0.6\text{B} + \text{BACE}}^{\text{MHA-RAG}} = 75.08$.
> >
> >
> > > **(Q2.) “Figure 1 shows that MHA-RAG freeze foundation model and only train the encoder the multi-head attention. Can you clarify in Training Details (line 309), what does the Models refer to?”**
> >
> > Yes, similar to other baselines, MHA-RAG freezes the foundation model and trains only the encoder along with its respective Projector or multi-head attention module. The term Models in line 309 refers to these trainable components including the encoder and multi-head attention based projector, with different foundation models (Qwen3-0.6B, Qwen3-4B, Llama3.2-3B-Instruct) serving as the frozen backbones. We will revise the text in line 309 to clarify this.
> >
> >
> > > **(Q3.) “What is the metric for evaluating PubMedQA dataset in Performance Metrics?”**
> >
> > The PubMedQA dataset provides categorical golden answers, primarily of the form yes or no. For example:
> > Q: Are seizure frequency and duration of epilepsy risk factors for postoperative seizure outcome in patients with hippocampal sclerosis?
> > A: Clinical factors such as seizure frequency and duration of epilepsy are not risk factors for postoperative seizure recurrence.
> > Accordingly, we evaluate model performance on PubMedQA using Effective Accuracy.
> >
> > > **(Q4.) “Missing Hyperparameter Details (Table 3): While Table 3 studies the effect of varying K, the corresponding value of H used in these experiments is not provided/What is the number of attention heads in Table 3?”**
> >
> > We consider number of heads as hyper-param and do a sweep over it in the range 1,2,4,8,16. Here are the best performing ones for K=1,5,10. We will add these details in the Appendix.
> >
> > |Qwen3-4B|K=1|K=5|K=10|
> > |:---:|:--:|:--:|:---:|
> > |BACE|1|2|4|
> > |BBBP|2|8|16|
> > |ClinTox|1|4|8|
> > |PubMedQA|2|8|8|
> >
> > |Llama3.2-3B-Instruct|K=1|K=5|K=10|
> > |:---:|:--:|:--:|:---:|
> > |BACE|1|2|16|
> > |BBBP|1|8|8|
> > |ClinTox|4|8|16|
> > |PubMedQA|1|8|8|

---

### Meta-Review · Area_Chair_ii7K · 2026-01-09

**Summary:**

The proposed method is interesting, and the authors' rebuttal was responsive. However, the concerns regarding the rigor of the efficiency evaluation, and the need for broader comparisons remain substantial. The authors are encouraged to incorporate the rebuttal experiments into the main paper and rigorously address the efficiency measurement issues for a future submission.

**Reviewer Concerns:**

All the reviewers raised concerns regarding the empirical evaluation of the proposed methods, such as Insufficient and Incomplete Baseline Comparisons, Limited Scope and Generalization of Benchmarks, and issues with efficiency evaluation. Some reviewers also question the robustness of the proposed method and the scalability concerns.

**Reviewer Scores:**

All the reviewers vote for rejects with ratings 4, 4, 2, 2, and both 2 have high confidence. Also, the concerns from the reviwers are not fully addressed in the rebuttal, so it's likely that the two reviewers with rating 2 won't change their decision.

---

### Decision · Program_Chairs · 2026-01-26

Reject